# “It’s All Just Marketing”, a Qualitative Analysis of Consumer Perceptions and Understandings of Nutrition Content and Health Claims in New Zealand

**DOI:** 10.3390/ijerph19063510

**Published:** 2022-03-16

**Authors:** Lucy Stuthridge, Donnell Alexander, Maria Stubbe, Paul Eme, Claire Smith

**Affiliations:** 1Department of Human Nutrition, University of Otago, Dunedin 9016, New Zealand; lucy.stuthridge@gmail.com; 2New Zealand Food Safety, Ministry for Primary Industries, Wellington 6140, New Zealand; donnell.alexander@mpi.govt.nz (D.A.); paul.eme@mpi.govt.nz (P.E.); 3Department of General Practice, University of Otago, Wellington 6242, New Zealand; maria.stubbe@otago.ac.nz

**Keywords:** nutrition claim, health claim, food labelling, qualitative, consumer

## Abstract

Nutrition content and health claims are widely used globally on both food labels and in food advertising. This study explored how New Zealand consumers understand, perceive, and use nutrition content and health claims on food labels. A qualitative approach was used with semi-structured in-depth online interviews and in-person focus groups including 49 participants, aged ≥25 years responsible for household food shopping. Transcripts were analysed using reflexive thematic analysis using inductive coding, with development of five themes—(1) aware of claims but did not use, (2) mistrust and scepticism, (3) confusion and misinterpretation, (4) using claims to guide food choice, and (5) not all claims are equal. For theme 1, price and habit were found to be the most influential in driving food choice. Underlying theme 2 was the perception by most of nutrition and health claims as marketing. Scepticism was exacerbated when nutrient claims were displayed on inherently unhealthy products. However participants with specific dietary requirements did find claims helpful. Restricting nutrient claims to foods meeting a healthy nutrient profile aligned to the existing Health Star Rating system, education about regulation and supporting claims with more contextual information may increase trust, the perceived value of claims and therefore their utility.

## 1. Introduction

Nutrition and health claims are messages typically on the front of food packaging. They often sit alongside other front-of-pack labels aiming to provide a quick reference for consumers to make a judgement on the relative healthfulness of a food [1]. Examples of front-of-pack labels include health logos (Singapore and Thailand), traffic light systems (e.g., the UK and Korea), warning labels (Chile and Finland) and health star ratings (NZ and Australia) [1]. As food systems become more connected globally with increases in trade in particular of cheap less healthy food, there is the need to consider ways to help consumers make informed food choices [2]. Clear food labelling is one strategy to both increase purchasing of healthy food and to encourage reformulation [3]. It is recommended as part of a range of public health policies to encourage consumers to select and purchase healthier foods according to the World Cancer Research Fund [4].

Nutrition content and health claims are voluntary statements on labels and are used as a marketing tool to emphasise the product’s positive attributes [5]. Nutrition content claims refer to the content of a nutrient or substance in a food, such as “a good source of calcium”, whereas health claims refer to a relationship between a food and a claimed health benefit, for example “fibre keeps your regular” [5]. The regulation of health and nutrition content claims varies between countries [6]. Some countries do not allow health claims, including Nigeria, South Africa and Thailand [7]. In Europe, regulations permit nutrition content claims and health claims [5,8]. Health claims in New Zealand and Australia can either be ‘general’ or ‘high level’. ‘General level’ reflects a claim related to the maintenance of general health and well-being. In contrast, ‘high level’ signifies a food product that can help reduce the risk of a serious disease or a biomarker of a serious disease [9].

Since 2013 in New Zealand and Australia, nutrition content and health claims have been regulated under Standard 1.2.7 of the Food Standards Code [10]. There has been a considerable amount of effort invested in developing and enforcing the Standard but there is no knowledge of how effective the Standard is at conveying nutrition and health information to New Zealand consumers. Foods must undergo nutrition profiling to display a health claim, which ensures that they meet a minimal standard for nutrition density. However, nutrition content claims do not require any nutritional profiling other than confirming that they meet the minimal level for the specific nutrient claimed [11].

Nutrition content and health claims are used globally. The prevalence of nutrition content and health claims on foods varies by country but ranges from 26% to 50% of products [12,13]. Nutrition content claims are more common than health claims; and in 2017, over half (56%) of packaged foods surveyed in New Zealand displayed a nutrition content claim, and 2.7% displayed general-level health claims [14].

Nutrition content and health claims may assist shoppers with making informed decisions about their purchases and consumption of food and beverages, potentially leading to an improvement in diet quality [15]. However, some research has shown that nutrition content and health claims may increase purchasing and consumption of less healthy foods by creating a “health halo effect”, which increases consumer perception of the healthfulness of the food [16,17,18,19,20].

In contrast, research in the United Kingdom and Europe has shown that perceived healthfulness is not always increased [21], and some consumers mistrust health claims in particular [22]. Furthermore, comparing participants from different countries showed variations in the interpretation of nutrition claims as indicators of healthiness [21]. Qualitative research in Australia, with whom New Zealand share their s food regulations, found that consumers viewed nutrition content claims as misleading and were sceptical about all label claims [23,24]. Furthermore, the addition of a nutrition content or health claim on packaged food did not increase the likelihood of consumers picking healthy products but did increase the likelihood of choosing unhealthy products [25]. Adults with type 2 diabetes and pre-diabetes in New Zealand reported health claims inhibited their ability to properly evaluate a product and rather perceived claims as marketing messages aiming to increase sales [26]. Consequently, the potential use of nutrition and health claims as information to educate consumers and enable the selection of healthier foods is being missed.

Food packaging and labels play a role in influencing food choice, but consumer food choice happens in a complex and multifactorial environment [27,28]. Food packaging may influence decisions at the point of sale; however, other factors including food policy, media, social and cultural environment, price, personal preferences, and food availability in the home play a significant role in food choice [29]. In addition, there is an ethical debate about whether the food industry should be able to market food products on their healthfulness [30]. At the heart of this debate is where responsibility lies: with the consumer to make the correct decision, or with the manufacturer to provide transparent and accurate information and guidance? With the widespread use of claims on food labels in New Zealand and conflicting literature on how consumers perceive claims, there is a need to better understand consumer perspectives regarding nutrition content and health claims. Therefore, this research study aimed to explore how New Zealand consumers understand, perceive, and use nutrition content and health claims on food labels, using a qualitative approach to allow in-depth exploration of consumers’ interactions with the food environment.

## 2. Materials and Methods

A critical realism framework was used to underpin the research methods and interpretation of the data. This approach seeks to understand fundamental mechanisms that influence reality [31] and fits with the aim of this research to unravel the underlying factors influencing people’s use, perception and understanding of nutrition content and health claims. The data analyses used a reflexive approach, with the researchers applying an interpretive lens and actively engaging with the data [32]. The first author (LS) collected all data using semi-structured interviews and focus groups and completed the initial analyses. This study received ethical approval from the University of Otago Human Research Ethics Committee (Ref D20/172).

### 2.1. Participants

Participants were recruited from Dunedin, Nelson and Wellington (two provincial centers and the capital city of New Zealand, respectively) through supermarket noticeboard posters and Facebook posts. To ensure a diverse range of participants were included, a purposeful sampling strategy was used. Participants were screened against the inclusion criteria, which included being over 25 years, being the primary food shopper in the household, and living in Dunedin, Wellington, or Nelson (Table 1). For eligible participants, further information was collected, including whether they had a Community Services Card (a measure of low-income participants), level of education, age, presence of children in the household and ethnic group. Participants were selected on a case-by-case basis to ensure maximum possible variation in the sample. Table 1 outlines the rationale behind the inclusion criteria and sampling frame and the number of participants we aimed to recruit. In total, 45 people completed the screening questionnaire for the interview, and 88 people completed it for the focus groups. Recruitment for the interviews and focus groups continued until there was sufficient representation of households with children, households with a Community Services Card and Māori and Pasifika households (see Table 1, which shows the target number of participants for the above demographics). A description of the characteristics of interview and focus group participants is provided in Table 2. Participants received a $30 supermarket voucher as acknowledgement for their time.

### 2.2. Interviews

Ten participants provided written consent and completed the in-depth semi-structured online interviews with LS (a postgraduate student) using Zoom conferencing between August and October 2020. The purpose of the interviews was to understand consumer knowledge and use of nutrition content and health claims before developing the question guide for the focus groups. Due to social distance and travel restrictions in New Zealand during the COVID-19 pandemic, the interviews were online. The interviews ranged from 30 to 45 min and were audio-recorded on a digital voice recorder (Endeavour, EN10).

The interview guide included nine topics: general shopping, food labels, awareness of nutrition content and health claims, use of nutrition content and health claims, interpretation of nutrition content and health claims, perceptions of nutrition content and health claims, knowledge of nutrition content and health claim regulations, understanding of different claims, and factors to increase consumer use of nutrition content and health claims (Table A1). The interview guide was pre-tested on two individuals who were not included in the results. One pre-test was in person and the other over Zoom conferencing. The interview guide also included two stimulus activities (Figure A1 and Figure A2), highlighted below in the stimulus activity section.

### 2.3. Focus Groups

Seven focus groups facilitated by LS were completed between October to December 2020. Three focus groups were held in Dunedin (late-October) with eight, six and three participants in each group; three focus groups in the Wellington region (mid-November), which included six, five and three participants each; and one focus group in the Nelson region (early-December) with seven participants. All participants provided written consent. A research assistant was present at each focus group to take notes, observe and assist with setting up the room, greeting and refreshments. The Dunedin focus groups and two Wellington focus groups were in research rooms on the Dunedin and Wellington University of Otago campuses. The third Wellington focus group was in a private meeting room at the Te Rauparaha Arena in Porirua. The Nelson region focus group was in a meeting room at the Motueka Community House. The focus groups took 60 to 90 min and were audio-recorded on two devices (Endeavour EN10 and a mobile phone).

The focus group question guide (Table A2) was developed from the initial coding of the interviews and the literature review. Initial analysis of the interviews (by LS) showed that participants were familiar with nutrition content and health claims, but few were not consciously using them. Therefore, the focus group included three stimulus activities to explore further participant interpretation and understanding of nutrition content and health claims (Figure A3).

### 2.4. Stimulus Activities

Activities within focus groups can offer an enjoyable and productive supplement to the questions to generate rapport and stimulate discussion [37]. The semi-structured in-depth interviews included two activities and the focus groups three activities. Visual aids were shown to participants as examples of food product packaging with nutrition content and health claims. These were selected to be familiar foods typically consumed in a New Zealand diet. The activities aimed to stimulate discussion rather than determine the participants’ ability to identify healthy products.

In the semi-structured in-depth interviews, the first activity displayed a selection of food packaging on the computer screen including three nutrition content claims and four health claims (two high-level and two general-level health claims) (Figure A1). The participants were asked questions after viewing the products regarding how they interpreted the claims and if the claim would impact their food choice (Figure A1). In the second activity, two snack bars were compared, one displaying nutrition content claims and one without. Participants were asked to identify any differences between the two products and which product they preferred.

The three focus group stimulus activities included the were: (i) rating four products with nutrition content and health claims on the front on a scale from healthy to not healthy and discussing the reasoning behind each rating; (ii) comparing the difference between two nutrition content claims on two breakfast cereal packets (one ‘very high in fibre’ and one ‘source of fibre’); and (iii) a comparison between two products with the question ‘which one you are more likely to purchase?’ (example one: two bars one with a claim and one without one; example 2: two breakfast cereal packets, one with a claim and one without) (Figure A3).

### 2.5. Data Analysis

One of the audio-recorded interviews was transcribed verbatim by LS and the remaining interviews and focus groups by a professional transcriber. Any information which might identify participants was removed from the transcriptions. Participants were offered the opportunity to review transcriptions and suggest changes. No participants requested this.

The software NVivo (version 12.0.0, QSR International Pty Ltd., Melbourne, Australia.) was used to assist with the analyses using initially a deductive and then an inductive approach. The initial analysis was completed by LS. The six-phase process of reflexive thematic analysis [38] was followed, with transcripts from the interviews and the focus groups combined. LS listened to the audio recordings and read the transcriptions to generate a list of initial codes using the research objectives as a framework. Quotes from the transcripts were allocated to the initial codes, and at the same time, new codes (and sub-codes) were created using an inductive approach. Throughout this process, codes were reviewed and discussed with CS. LS searched for themes amongst the codes, compiled all relevant codes to one theme, and created a thematic map of the different themes and subthemes. These themes were discussed with two other researchers (CS and MS), rechecked, and refined. The interview and focus group transcripts were analysed until data saturation was reached—data saturation is where no additional data are found in a dataset.

Previous research has shown that people with health conditions or shopping for others with health conditions [39], older adults [40], households with children and households with limited resources [41] might perceive and use nutrition content and health claims differently. To explore this aspect specifically, we undertook analyses of positive sentiment (i.e., identifying all codes which framed nutrition content and health claims positively). The frequency of making a positive statement about a nutrition content and health claim was assessed across the demographic variables age, households with children, participants with a Community Services Card and participants who reported a health condition or shopping for someone with a health condition (derived from the transcripts).

The lead author LS was a female postgraduate student who completed this project as part of a Master of Science in Human Nutrition. LS kept a reflective diary throughout data collection and analysis. She did not previously have specialised knowledge of food labels but had an undergraduate degree nutrition. LS had an existing relationship with one interview participant and three focus group participants. These participants were recruited similarly to the other participants and were selected based on the inclusion criteria. On the other hand, the prior relationship may have positively impacted the ease of discussion due to pre-established rapport. CS assisted in analysing the data by reviewing the codes and themes collated and providing feedback and suggestions. The Consolidated Criteria for Reporting Qualitative Research (COREQ) [42], a 32-item checklist, was used in designing the methods, analysing the data and reporting the results of this study (Appendix A).

## 3. Results

Ten adults participated in the semi-structured in-depth interviews and 39 in the focus groups with an age range of 26–69 years. Table 2 displays the demographic characteristics of the participants. Most were female (80%) and 60% were of New Zealand European ethnicity. One-fifth had a Community Services Card, and 40% had children living at home. Eight participants were identified who shopped primarily for health reasons related to food intolerances and allergies (6), diabetes (2), and weight loss (1).

### 3.1. Themes

Thematic analysis of the transcripts identified five major themes and eight subthemes (Figure 1). The themes, described in detail below with illustrative quotations, are: (1) aware of claims but did not use, (2) mistrust and scepticism of nutrition content and health claims, (3) confusion and misinterpretation of nutrition content and health claims, (4) using claims to guide food choice, and (5) not all claims are equal.

The first theme explores awareness of claims but highlights wider contextual factors that influence food choice decisions. So, although awareness and recognition are apparent, the influence of claims on purchasing decisions is not strong. The second theme relates to participants’ perception of the claims as marketing which links to the third theme where participants reported confusion in the interpretation of the claims, especially when considered with other parts of the food packaging and nature of the food. The fourth claim describes the way some participants use claims to guide food purchasing decisions. Finally, the fifth theme relates to how consumers view the importance of some claims compared to others.

#### 3.1.1. Theme 1: Aware of Claims but Did Not Use

Despite a general awareness of nutrition content and health claims, participants reported a wide range of factors influencing food shopping at the point of sale, along with broader contextual factors. Nutrition content and health claims were perceived as being relatively unimportant in influencing purchasing decisions compared to these other factors.

Claim awareness. Many participants reported that they were aware of and had seen nutrition content and health claims displayed on food packets. As one participant commented:
“Ay yeah, it’s hard not to notice if they’re all plastered all over the packaging”(Female, 46–60 years, participant 48)

They were also able to recall a large variety of different claims; however, nutrition content claims relating to sugar and fat were the most commonly recalled. Below, a participant highlights how they notice fat-free claims.

“I definitely notice the fat-free cause I don’t do fat-free.”(Female, 36–45 years, participant 45)

Dairy products, bread, juices, stock, potato chips and snacks, sweets, breakfast cereals, muesli bars, baby foods and margarine were all listed as foods that frequently displayed claims. In particular, breakfast cereals were noted as having the largest amount of information on their packaging.

Other types of nutrition-related food labels such as the Health Star Ratings and the Heart Foundation’s Pick the Tick (which is no longer being used on food labels in New Zealand) were commonly also recalled by participants. However, despite a general awareness of nutrition content and health claims on food labels, participants did not generally use or read them when shopping, with other factors having a greater influence on food purchasing behaviour. Talking about this, one participant said:
“I’ll see them and I’ll maybe look at them but I don’t really care.”(Female, 46–60 years, participant 25)

In contrast, the participants shopping for foods suitable for those with food allergies and intolerances were specifically looking for “free from” statements on the front of packs. One participant said:
“I, yeah, I mean I do look out for gluten-free and ah sometimes I do check, I’m often checking the detail because sometimes products will change and I have had products that have um, have changed, have been gluten-free and then…”(Female, 46–60 years, participant 6)

Price and budgeting. The most important influence on product choice was the price of food. Many participants considered this to be the main factor in deciding on one food over another. The comments below illustrate this:
“[with] all these claims and all this labelling, I’m still going to be bound by my budget. I have a family to feed, we’re on a very low income. Doesn’t matter what’s out there, I’m going to buy the cheapest, so I mean, it’s sad, but it’s the truth.”(Female, 36–45 years, participant 47)
“When I’m actually shopping, price is a big, big part of it”(Female, 46–60 years, participant 25)

Shopping for specials and searching for bargains repeatedly emerged as critical food shopping strategies. Additionally, participants were more likely to try a new product if it was on special, and some did not buy products unless they were discounted.

“If it’s not on special, I won’t buy it, just wait. You don’t need it”.(Female, 61+ years, participant 30)

Habits and preferences. When choosing food items, familiarity was important. Participants were more likely to buy something they had already tried before, as they trusted the brand and knew that other household members liked the product. Purchasing the same foods and brands was identified as an easy way to shop, especially when considering time constraints as expressed by two participants.

“So, you definitely get stuck in your ways of buying and shopping. Someone else needs to do the shopping every now and then introduce you to some more products.”(Female, 36–45 years, participant 36)

“Habit definitely, it’s a big one. Like oh yeah, that is, you know like where your hand just goes”(Female, 36–45 years, participant 43)

Choosing familiar brands and trust in these brands were important in food choice, with participants naming specific brands that they trusted and preferred to buy.

“I spend time looking at, um, comparing different brands. And then, once I’ve decided on a brand, I’ll stick with that one.”(Female, 25–35 years, participant 19)

Credence attributes. In contrast to making solely price-based decisions, some participants also considered other aspects of a product including country of origin and supporting local producers.

“Products made in New Zealand is a big factor as well. I don’t really want anything [made] overseas, products you know.”(Female, 25–35 years, participant 15)

Buying New Zealand-grown and -produced food seemed to be more important for fresh food such as fruit and vegetables, fish and seafood and meat.

“Obviously price as well does come into it but not if I looked at a packet of prawns or salmon and I see that it was an overseas product.”(Female, 35–45 years, participant 36)

Another attribute considered when choosing food products was the packaging. Some participants valued a sustainable approach and checked for sustainable options, for example, ‘eco’ packaging or packaging portrayed as ‘natural’ and ‘environmentally friendly’, as expressed by two participants.

“I think I sometimes get pulled on the ‘eco’ thing.”(Female, 25–35 years, participant 33)

“I would buy [the natural looking one] because you know, it’s [packaging/branding] played my mind, playing on my, wanting to be natural...... Be a good mummy cause I’m buying something, you know, from the earth.”(Female, 36–45 years, participant 47)

Cultural factors. Māori participants highlighted the importance of food in providing hospitality. This participant describes how her food choice was influenced from a young age by her culture and upbringing. This shows that there are many factors beyond food labelling that influence food choice, such as hospitality.

“Coming from a family with quite large women and like especially on my Māori side, I remember going, and the whole theory was oh you’re so little, quick sit down, I’ll give you some more, and that whole hospitality thing, feed, feed, feed, feed, and my family are feeders, we are all feeders, we are still feeders and I mean, and we’re all big, all of us.”(Female, 36–45 years, participant 43)

Manaakitanga is a traditional value that is hugely important in Māori culture—it is providing hospitality and generosity for others [43], as described by one participant.

“Food for us was a manaaki sharing, giving, loving.”(Female, 51–60 years, participant 42)

#### 3.1.2. Theme 2: Mistrust and Scepticism

This theme describes the general mistrust that participants had relating to nutrition content and health claims, although this mistrust was stronger in relation to nutrition content claims than health claims. A claim was an indication that something was being “hidden”. Three subthemes were identified: the perception of nutrition content and health claims as mainly marketing messages, the view that some claims were of little relevance or “meaningless”, and the perception that they were mainly unregulated.

Marketing and advertising. Participants perceived claims as marketing messages designed to “hook you in” and not a legitimate source of information. Participants also expressed their own sense of empowerment and control by not being coerced to purchase a product just because it had a claim, and checking the back of the packet (presumably for ingredients and nutrition information) rather than simply believing the claims. For example, one participant said:

“I think it’s marketing, a lot and that’s what really pees me off to see it. It makes me think, oh they think I’m stupid.”(Female, 36–45 years, participant 44)

In reference to using other information, another participant said:
“No, I take it as advertising, so I don’t believe it. I would rather read the label myself [back of packet]. It [the claim] might be a hook for me to read the label [back of packet], but it wouldn’t make me buy it.”(Male, 46–60 years, participant 2)

Some participants perceived nutrition content and health claims as a way of covering up less desirable features of a product, such as more processing or being high in another nutrient such as fat, salt or sugar. When asked about the types of foods that displayed nutrition and health claims, one participant reported:
“Processed stuff. Because they’re trying to justify; like when I see something like that, I’m like, it kind of makes me concerned.”(Female, 25–35 years, participant 33)

Participants were particularly sceptical about the sugar content of foods with nutrition content claims, especially for those with ‘reduced fat’ claims.

“I’m always like, hmm, no I won’t buy the fat-free because of what I’ve seen about how they just add sugar to make it taste better. I’d rather have the full fat.”(Female, 25–35 years, participant 18)

Meaningless messages. Many participants believed that some of the nutrition content and health claims were pointless and did not provide any new information. Examples included dairy products with claims relating to calcium and protein. This focus group participant refers to the claims displayed on one of the bars in Activity 3.

“I just think any bars with nuts in them, that they have got fibre, they have got protein. They’ve got nuts there, that’s what nuts are.”(Female, 46–60 years, participant 41)

Unaware of regulation. There were mixed results for participants’ understanding of how claims are regulated. Half the participants thought that all food manufacturers could display a nutrition content or health claim if they desired and believed the regulations around nutrition content and health claims were very relaxed or non-existent. One participant said:
“My understanding is that there’s not a lot of directives as to what you can or can’t put on a label”(Female, 46–60 years, participant 8)

The other half agreed that there must be some regulation around claim use on food labels although no one reported specifics. One individual stated the following:
“I think that a lot of people would buy them and trust the company to have done the testing and the research behind the claim, whether or not that might be true or not, but it must be unlawful for them to do it without it.”(Female, 35–45 years, participant 48)

#### 3.1.3. Theme 3: Confusion and Misinterpretation

The focus group activities where participants had to rank products as healthy and less healthy and compare products with different claims revealed much confusion and misinterpretation of food labels, including nutrition information. This theme is closely related to theme 2, with confusion and misinterpretation further exacerbating mistrust and scepticism. The nutrition content claims that were the most difficult to interpret were ‘source of x’, and nutrition content claims that specified an amount. For example, ‘contains 13 g of protein per serve’. Talking about this issue, a participant said:
“13 g per serve—what does that mean?”(Female, 46–60 years, participant 28)

The comment below illustrates how some participants perceived that there was inadequate information and context provided for the claim to be interpreted.

“56% less [sugar] or whatever. But how much is in the other stuff?”(Male, 61+ years, participant 29)

If there were several different types of claims on the food packaging, then confusion was further heightened. Participants also demonstrated confusion when there were other elements on the food label such as the Health Star Rating. For example, if a food displayed only two health stars but had a nutrition content claim, this was confusing. The difficulty in being faced with multiple sources of information is expressed by one participant below.

“[I am] .... overwhelmed because there’s so many different things on the box.”(Female, 25–35 years, participant 3)

#### 3.1.4. Theme 4: Guiding Food Choice

Several factors seemed to influence whether participants used nutrition content and health claims to make purchasing decisions. Two participants reported having type 2 diabetes and two had allergies and food intolerances. Four participants were shopping for others who required gluten-free foods. Participants with allergies or intolerances or those shopping for family members with allergies and intolerances appreciated having easy-to-view information. For example, one participant said:
“Well, one thing I noticed about this is it says gluten-free, which saves you having to look at the back.”(Female, 46–60 years, participant 17)

Another participant used the presence of nutrition content claims to compare two products, and this is described by one participant below.

“Might use it [the claim] to make a comparison between two brands that I haven’t tried before. Look at their claims on the front before I flip it around and read the labels”(Female, 46–60 years, participant 22)

A small group of participants reported that although they did not currently use nutrition content and health claims, they had previously paid attention to them, such as when they were shopping for children or on a weight loss diet.

The sub-analysis outlined in Table 3, examining positive statements based on selected demographic variables, showed that participants under the age of 40 years, those without children, and those with health problems were more inclined to have a positive attitude towards nutrition content and health claims than older participants and parents.

#### 3.1.5. Theme 5: Not All Claims Are Equal

Some claims were interpreted as being more legitimate and trustworthy than others. For example, if the product was locally produced, then allergen labelling was considered more credible. The wording of the claims also influenced participant interpretation of credibility. For instance, nutrition content claims that stated ‘high in’ rather than ‘source of’ were correctly interpreted as having more of the nutrient or ingredient in the claim, as illustrated in the comments below.

“I said source just indicates there’s a little drip in there. It’s not real, it’s not a grunty amount of good fibre.”(Female, 46–60 years, participant 25)

“I think, ‘source of fibre’ is pretty vague, because … technically, it has fibre in it, whereas ‘high in fibre’, to me, is saying, “I’m higher than the average.”(Female, 35–45 years, participant 27)

In considering the example of a health claim, many participants interpreted ‘clinically proven’, as more trustworthy and credible, indicating that health claims may be more believable than nutrition content claims. One participant linked the health claim to research:
“Clinically proven… which means … they have done their research. So, even, here it says, ‘part of a healthy diet’, still, they are committed to do something, so, they have done some work. So, more likely, I will try it and give it a go.”(Female, 25–35 years, participant 19)

### 3.2. Relationship between The Themes

Five themes were identified and Figure 1 proposes how the themes and subthemes relate. The subthemes “claim awareness and habits and preferences” (Theme 1) relate to “using claims to guide food choice” (Theme 4). Participants were aware of nutrition content claims, and those with a health condition and/or a dietary requirement were using specific nutrition content and health claims. Confusion and misinterpretation of nutrition content and health claims (Theme 3) by participants led to increased mistrust and scepticism (Theme 2). Some claims were better understood than others (Theme 3), but were also perceived as more credible, thus linking to the theme “not all claims are equal” (Theme 5). In addition, the subthemes marketing and advertising and meaningless messages (Theme 2) reiterate the fact that not all claims were perceived as equal (Theme 5), as some claims were not trusted (fat-free claims) whereas others were thought to be meaningless by stating facts that were already widely known (protein claims on snack bars containing nuts). Lastly, Theme 5 is linked in with Theme 3 as people found some nutrition content claims to be confusing (per gram amount of protein per serve) while they considered others to be clear (high in fibre) and health claims to have research behind them.

## 4. Discussion

Food Standards Australia and New Zealand (FSANZ) introduced Standard 1.2.7 in 2013 to provide rigour and standardisation of nutrition content and health claims in Australia and New Zealand. Since the implementation of the Standard, there has been no research on the value of nutrition content and health claims from a consumer perspective in New Zealand. This study provides insight into how consumers understand, perceive and use nutrition content and health claims on food labels. Five key themes were identified—aware but did not use, mistrust and scepticism of nutrition content and health claims, confusion with the interpretation of nutrition content and health claims, and using claims to guide food choice, and not all claims are equal.

Despite the abundance of nutrition content and health claims on food labels, most consumers perceived these as a marketing tool and did not report consciously using them in food purchasing decisions. In addition, confusion due to difficulties interpreting the claims with other aspects of the food label, such as nutrition information panels and health star ratings, combined with limited knowledge of the regulation of claims, exacerbated mistrust and scepticism.

The qualitative approach allowed for an exploration of the broader contextual factors influencing the use of nutrition content and health claims and food purchasing decisions. Although participants reported awareness and recognition of nutrition content and health claims, habits and price were the most salient factors influencing food purchasing decisions. This supports previous research that consumers purchase food mainly out of habit [44,45,46] and are reluctant to purchase new foods [39]. In addition, Aschemann and Hamm found that the choice of products that displayed claims was negatively impacted by brand loyalty [17]. Congruent with other research, cost and participants’ budgets were also major determinants of food choice [28,41,47].

An important contextual finding was that in Māori culture, food choice and eating are influenced by social factors such as ‘manaakitanga’ (providing and receiving hospitality and generosity) [43]. Manaakitanga is important in Māori culture. The importance of ‘feeding’ and receiving is more important than choosing healthy food options as it is equally essential not to refuse the hospitality given by others [43]. This finding highlights the importance of the cultural role of food and how food is linked to identity, and how this may override health-related messages such as nutrition content and health claims.

An important finding was the mistrust and scepticism participants had of the motives behind manufacturers using nutrition content and health claims on food labels. This is again consistent with other research, showing consumers tend to view claims with scepticism [19,23,26,48,49,50,51]. Many participants questioned the accuracy of the claims, often based on a negative experience. Other research in Australia, France and Sweden also found that consumers lacked trust in nutrition content and health claims [48,49]. Paradoxically, despite the lack of trust and scepticism reported in this study and others, a recent systematic review has shown that nutrition content and health claims still increase the likelihood of a consumers’ purchasing a food [16].

The mistrust and scepticism in claims reported in our study could be due to the reported lack of understanding of the claims’ meanings. Many focus group and interview participants found claims challenging to understand and confusing; this was particularly the case for nutrition content claims. Other research found that consumers preferred more straightforward claims such as nutrition content claims [49,52], and New Zealand research also supports this [53]. Possibly this indicates a need for greater transparency because of the contradicting information from nutritional influencers such as friends and family members, the media, food industry, social media, and Government. Conflicting information can lead to confusion about what to believe [54,55,56].

Although this and other research [39,49] have found that nutrition content and health claims were not a key driver of food choice, in this study, it was established that claims are helpful for people who have specific dietary requirements and health issues. This is supported by two international studies that found participants with a health concern were more likely to use nutrition content and health claims [39,49].

Though many participants perceived nutrition content and health claims in a sceptical and mistrusting light, a few participants trusted and believed in the claims. However, the trust appeared to be greater for some claims than others. For example, the ‘high in fibre’ nutrition content claim was trusted more than the ‘source of fibre’ nutrition content claim, as participants viewed the ‘source of’ claim as unclear and confusing. This shows that the clarity of claims is crucial when conveying information on the front of packets. Additionally, it appeared that the participants trusted health claims more than nutrition content claims as they were perceived to be backed by research and evidence. One study based in Ireland found that consumers preferred nutrition content claims over health claims as they were easier to understand [52], whereas other research found that different health claims had varying degrees of credibility depending on the type of health claim [48]. The examples of health claims discussed in this study were not extensive enough to determine this.

### 4.1. Strengths and Limitations

This study was the first New Zealand qualitative study that specifically examined nutrition content and health claims from a consumer perspective since the introduction of Standard 1.2.7 in 2013. The participants had a diverse range of demographics which reflected the makeup of food shoppers in New Zealand. However, there was an underrepresentation of Pasifika people. We recommend further research using culturally relevant recruitment and research methods. A range of nutrition content and health claim examples were used in the semi-structured in-depth interviews, but only one health claim example was used in the activities, so it is more difficult to draw conclusions on attitudes towards health claims specifically, compared with nutrition content claims. However, far fewer health claims are used on food products than nutrition content claims in New Zealand [14]. Most of this research occurred outside the time of strict COVID-19 lockdowns in New Zealand, though even at such times New Zealanders were still able to leave their house to shop for food at a supermarket. There were some minor food shortages due to panic buying. The lockdown also influenced how people shopped for food because access to food changed, people were cooking more at home, were ‘panic buying’ certain items, they visited supermarkets less frequently, and shopped for food online where possible [57,58,59]. It is possible that people’s perceptions of food and food labelling might have been altered at this time, with factors such as credence and price becoming viewed as more important than before [58,59].

### 4.2. Implications

The focus of this research was nutrition and health claims; however, the research also highlights the complexity of food choice and the small role that food labels play as a component of decision making. The findings do not suggest that participants passively believe claims but instead use other cues to reach conclusions about the relative healthfulness of a food. As found in other qualitative research, the “health halo” effect was not apparent in this study [20]. Although participants were aware of the claims, the findings suggest that they were not a large component of the food choice decision process. Many participants were particularly sceptical of nutrition content claims, and cited examples such as “low in fat” claims potentially being untrustworthy on foods which are high in sugar. This issue may arise because nutrition content claims can be made on foods that meet the requirements for a specific nutrient without needing to meet broader nutrition criteria. For example, a product may be high in salt/fat and sugar or have a low health star rating, but still be able to make a “source of vitamin C” claim if it contains more than 10% of the recommended daily intake (RDI) for vitamin C in one serving. Other New Zealand research found that nutrition content claims were found on one-third of ‘less-healthy’ products and almost half of ‘healthy’ products [60].

To address these discrepancies, the criteria for making a nutrition claim could be extended to also including a health star rating above a specified amount. This would mean only healthier foods could display nutrition content claims.

In addition to this, further information could be provided on the food label to support the claim. For example, it is a requirement that if a nutrition content claim is made for a certain nutrient, the level of that nutrient must be listed in the nutrition information panel (NIP) on the label. However, further contextual information may be helpful. For example, a claim that states the product contains 13 g of protein per serve should also state that it is approximately 25% of an adult’s daily requirement for protein. This recommendation could ensure that the language of claims does not mislead the consumer. Additionally, participants reported that they found the wording ‘source of’ particularly confusing and vague. Standard 1.2.7 states that a product needs to contain at least 10% of the stated nutrient’s RDI per serve to use a “source of” statement; however, this is unclear to the consumer. A recommendation to address this confusion is to state that the nutrient or substance contributes to at least 10% of the recommended daily intake on the back of the packet or mandate the use of a %RDI column in the NIP (which is currently a voluntary option).

Lastly, half the participants believed that nutrition content and health claims were unregulated in New Zealand. A health promotional approach is recommended to inform consumers of the regulations behind the claims and the strict requirements that manufacturers must meet to display a claim. This approach is suggested to promote nutrition content and health claims as a reliable and trustworthy source of nutrition and health information, enabling consumers to make healthful food choices, rather than just viewing them as marketing messages that cannot be trusted. An example of a successful nutrition labelling campaign in New Zealand was the Heart Foundation’s Pick the Tick campaign. It influenced consumers and also affected food manufacturers to reduce the amount of fat and salt in foods to meet the eligibility criteria [61]. Similarly, promotion of the Health Star Rating system has had some positive results in terms of consumer understanding over time as reported in the 5 year review of the system [62]. Increasing trust and understanding of nutrition and health claims as informative and truthful messages rather than simply a marketing tool could mean the claims are of more value to people.

## 5. Conclusions

This study has presented qualitative findings regarding consumer understanding, perceptions, and use of nutrition content and health claims on food labels in New Zealand. Nutrition content and health claims were helpful for a minority subgroup of consumers with specific dietary requirements such as coeliac disease. However, nutrition content claims were perceived as mainly marketing messages, designed as a hook-in to sell a product, and did not appear to be valued by participants. There was little understanding of how claims are regulated and some confusion in interpreting claims, especially in the context of other label information. This further contributed to many perceiving nutrition content and health claims with scepticism and mistrust.

## Figures and Tables

**Figure 1 ijerph-19-03510-f001:**
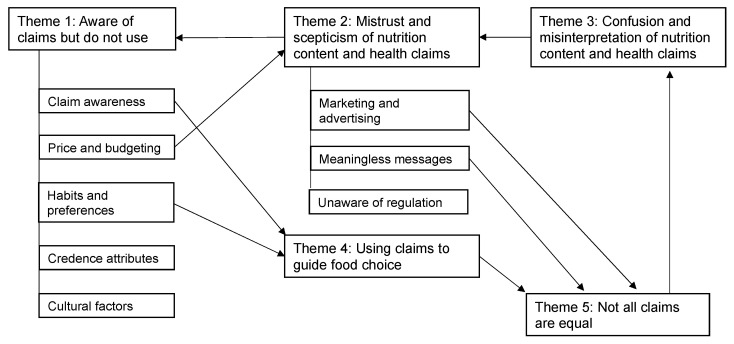
Thematic map.

**Table 1 ijerph-19-03510-t001:** Recruitment selection rationale.

Demographic Characteristics	Rationale	Target Number
*Inclusion criteria*		
Over 25 years	Younger adults shop differently than older adults with increased purchasing of takeaway and convenience foods [33].	N/A
Primary food shopper	Make food purchasing decisions for the households and are exposed to nutrition content and health claims more other food shoppers.	N/A
Geographical location	Dunedin, Nelson and Wellington were chosen as they provide a measure of geographical diversity (North and South Island, larger versus smaller urban center, and less and more ethnic diversity). Additionally, these locations were convenient as researchers were located in each of these cities.	N/A
*Purposeful sampling*		
Community Services Card	This allowed for the representation of participants with potential economic constraints on their food budget. In NZ, people are eligible to have a Community Services Card if they have a low income relative to their household size. A Community Services Card allows discounts on healthcare, travel and concessions on some services [34].	10
Participants with children	People with children might shop differently or consider different factors in their food purchasing decisions. It was important to include participants with children as 49% of New Zealand households contain children [35].	15
Ethnic group	The study aimed to include 14% Māori and 7% Pasifika participants. Māori represent 16.5% and Pacific peoples represent 8.1% of the New Zealand population [36].	Māori—6–7 Pasifika—3–4

**Table 2 ijerph-19-03510-t002:** Demographic characteristics of interview and focus group participants.

Demographics	Interview *n* (%)	Focus Groups *n* (%)
Age Categories (Years)		
25–35	3 (30)	15 (39)
36–45	0	7 (18)
46–60	5 (50)	14 (36)
61+	2 (20)	3 (8)
Gender		
Male	2 (20)	4 (10)
Female	8 (80)	35 (90)
Ethnicity		
New Zealand European	6 (60)	24 (61)
Māori	1 (10)	6 (15)
Pasifika	1 (10)	1 (3)
Other ^1^	2 (20)	8 (21)
Level of education		
No qualifications	1 (10)	4 (10)
NCEA L1	1 (10)	9 (23)
Trade	1 (10)	1 (3)
Bachelor’s degree	4 (40)	11 (28)
Postgraduate qualification	3 (30)	10 (26) 3 (8)
Children (<18 years) living at home	5 (50)	14 (36)
Community Services Card ^2^	2 (20)	9 (23)
City		
Wellington	5 (50)	17 (44)
Dunedin	3 (30)	15 (38)
Nelson	2 (20)	7 (18)

^1^ Other ethnic group includes Israeli, Malaysian, Middle Eastern, European, and American; ^2^ Holds a Community Services Card [34].

**Table 3 ijerph-19-03510-t003:** Sub-analysis: number of participants reporting positive codes relating to nutrition content and health claims by demographic characteristics.

Demographics	*n*	Proportion (95% CI)	*p*-Value
Age			
<40 (*n* = 22)	16	0.73 (0.54, 0.92)	
>40 (*n* = 27)	11	0.41 (0.22, 0.60)	0.025
Shopping for health problem			
Yes (*n* = 8)	6	0.75 (0.45, 1.05)	
No (*n* = 41)	21	0.51 (0.30, 0.720)	0.242
Community Services Card			
Yes (*n* = 11)	6	0.55 (0.15, 0.94)	
No (*n* = 38)	21	0.55 (0.34, 0.76)	1.00
Children			
Yes (*n* = 19)	7	0.36 (0.4, 0.71)	0.151
No (*n* = 30)	20	0.67 (0.46, 0.87)	

## Data Availability

The data presented in this study are available on request from the corresponding author.

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
