# Peer review of "“It’s All Just Marketing”, a Qualitative Analysis of Consumer Perceptions and Understandings of Nutrition Content and Health Claims in New Zealand"

_ijerph, 2022, doi:10.3390/ijerph19063510_

Round 1
Reviewer 1 Report
Thank you for allowing me to read this interesting manuscript. The findings are clearly presented and discussed. The introduction does provide a solid rationale for the study, but I feel that it would be useful to support this with a reasoning for why New Zealand in particular. However, overall I only have some minor comments that the authors may wish to consider.
Abstract
It would be beneficial if the results section of the abstract was supported with some data, rather than just the descriptions.
Introduction
Remove lines 30-38 as this is just the template guidance.
Lines 74-76: This is an interesting issue; Is there merit exploring the consequences of these findings, and how they relate to the present study?
Lines 87-89: Why is there an ethical debate, and what are the implications of this? I feel that this could be explored in order to strengthen the rationale of the present study.
Methods
Really clear description of the recruitment strategy.
Discussion
Overall, the results are clearly discussed. However, could the recommendations be developed in order to become more like practical implications e.g. how could nutritional content claims be made less confusing in a similar way there is an example of how the ‘pick the tick’ campaign was implemented?
I hope that the authors fine the above comments to be helpful and in the constructive manner they are intended.
Reviewer 2 Report
This is an interesting article. As we have known, it is worth investigating whether what we want consumers to understand is equivalent to what consumers actually receive. The research does make sense, both for businesses and consumers, as well as for those lawmakers and supervisors. The authors also conducted effective data collection and analyzed them in detail to provide us first-hand survey information. But in my opinion, as an investigative study, the amount of data in this paper is relatively small, which makes it difficult to carry out multi-dimensional analysis, so as to the result is superficial statements. For example, for well-known reasons, the consumer cognitive level is closely related to age, education level, and economic capacity. Only 49 valid data may be difficult to support such a cluster analysis.
In addition, the language of this article is not refined enough. For example, in the Introduction, Materials and Methods, and Discussion, sentences or paragraphs that are not relevant to this paper can be deleted. I tried to revise it, but the work is huge, so I gave up. Pls review it carefully and simplify the language. In the Results, it’s suggested the authors consider in what way the data are presented. I do not recommend stating the findings by quoting, as this is an academic manuscript rather than an investigative report.
Reviewer 3 Report
MDPI Peer Review…“Its all just marketing”
February 23, 2022
Comments and Suggestions for Authors
The first paragraph of the Introduction was a bit confusing as to whom it was directed. However, overall, a good introduction. Including descriptions of labels from different countries allows the reader to gain a quick insight into random patterns of packaged food labelling.
The introduction appropriately outlined the inadequacy of labelling globally. Perhaps a reminder of how widely travel and food exports have become, might have strengthened the need for useful labelling that provides appropriate nutritional information.
For the most part, the methods appear appropriate. The researchers attempted to be inclusive in respect to age, ethnicity, income, education level, as well as persons who were the most likely to be the shoppers.
While, it may have been difficult, the study would have been more representative of the New Zealand population, if the Pasifika sample was closer to the percentage in the population. For minority populations, especially those who are so few, it is very important because often, they are the ones that are likely to be most ill-affected.
The visual aids were excellent stimulus activities to elucidate unedited responses. There was good explanation of categories, e.g. what is meant by “high level”. This allows the reader to gain confidence that the participants were somewhat “standardized” into the language used in the study.
Providing feedback of how participants made food choices is useful. The fact that shoppers are somewhat skeptic of labels, prefer to purchase “home-made and culturally familiar foods ” and items with which they are familiar. It appears that labelling with no “nutritional profiling” and missing nutrient levels did not appeal to be of “added value” to the participants. In fact one comment is that, labels are viewed as a “hook” to the shopper. Perhaps these comments should be highlighted in the conclusions.
A good and useful study, with some minor issues, such as the low percentage of Pasifika among the participants and study group.
Round 2
Reviewer 2 Report
Thanks for the authors' reply. The author had answered my questions and made appropriate verification. In my opinion, this manuscript is suitable for publication.